# Expression of the Immunohistochemical Markers CK5, CD117, and EGFR in Molecular Subtypes of Breast Cancer Correlated with Prognosis

**DOI:** 10.3390/diagnostics13030372

**Published:** 2023-01-19

**Authors:** Carla E. Schulmeyer, Peter A. Fasching, Lothar Häberle, Julia Meyer, Michael Schneider, David Wachter, Matthias Ruebner, Patrik Pöschke, Matthias W. Beckmann, Arndt Hartmann, Ramona Erber, Paul Gass

**Affiliations:** 1Department of Gynecology and Obstetrics, Erlangen University Hospital, Comprehensive Cancer Center Erlangen-EMN, Friedrich Alexander University of Erlangen–Nuremberg, 91054 Erlangen, Germany; 2Würzburg University Hospital, Institut für Pathologie, Julius-Maximilians-Universität Würzburg, 97070 Würzburg, Germany; 3Institute of Pathology, Erlangen University Hospital, Comprehensive Cancer Center Erlangen-EMN, Friedrich Alexander University of Erlangen–Nuremberg, 91054 Erlangen, Germany; 4Institute of Pathology, Weiden Hospital, Weiden in der Oberpfalz, 92637 Weiden in der Oberpfalz, Germany

**Keywords:** early breast cancer, therapy, prognosis, CK5, CD117, EGFR, triple-negative breast cancer

## Abstract

Molecular-based subclassifications of breast cancer are important for identifying treatment options and stratifying the prognosis in breast cancer. This study aimed to assess the prognosis relative to disease-free survival (DFS) and overall survival (OS) in patients with triple-negative breast cancer (TNBC) and other subtypes, using a biomarker panel including cytokeratin 5 (CK5), cluster of differentiation 117 (CD117), and epidermal growth factor receptor (EGFR). This cohort–case study included histologically confirmed breast carcinomas as cohort arm. From a total of 894 patients, 572 patients with early breast cancer, sufficient clinical data, and archived tumor tissue were included. Using the immunohistochemical markers CK5, CD117, and EGFR, two subgroups were formed: one with all three biomarkers negative (TBN) and one with at least one of those three biomarkers positive (non-TBN). There were significant differences between the two biomarker subgroups (TBN versus non-TBN) in TNBC for DFS (*p* = 0.04) and OS (*p* = 0.02), with higher survival rates (DFS and OS) in the non-TBN subgroup. In this study, we found the non-TBN subgroup of TNBC lesions with at least one positive biomarker of CK5, CD117, and/or EGFR, to be associated with longer DFS and OS.

## 1. Introduction

In the treatment of breast cancer, it is crucially important for clinicians to have information about the likely prognosis and factors capable of predicting the patient’s response to therapy. This makes it possible to design individual risk profiles for each patient, as an aid in decision-making about whether the patient will be able to benefit from a specific treatment. It also helps to identify patients who will not benefit from therapy, or are unlikely to achieve remission, due to prognostic factors—thus avoiding unnecessary treatments [1]. A predictive as well as a prognostic factor can allow assessment of both the probability of a response to a specific cancer therapy and will also produce statements about the prognosis even without therapy [2].

Special molecular subtypes of early breast cancer are preferably treated with neoadjuvant chemotherapy [3,4,5]. Neoadjuvant systemic therapy may result in complete eradication of invasive cancer in the breast and axillary lymph nodes, which is defined as pathological complete response (pCR). A pCR after neoadjuvant chemotherapy is associated with a longer event-free survival (EFS)–the length of time after primary treatment the patient remains free of disease recurrence–and overall survival (OS) [6,7]. As breast cancer treatment becomes increasingly personalized, artificial intelligence can allow early prediction of pCR through Magnetic Resonance Imaging (MRI) [8], and a hierarchical clustering procedure which offers valuable additional information about data-driven individualized therapies [9,10].

Triple-negative breast cancer (TNBC) is associated with poorer outcomes and a higher risk of a distant recurrence in comparison with other molecular subtypes of breast cancer [11]. However, the OS and EFS rates are higher in TNBC patients with pCR than in those without pCR [12]. The effect of a pCR on the prognosis is independent of the treatment used [13].

Clinical and pathological factors (e.g., pT, pN, tumor grade, and proliferation rate using Ki-67 expression levels) are clinically well-established parameters for predicting the prognosis and treatment response. Molecular profiling, epigenetics and the quest for individualized diagnostics and treatment of breast cancer are becoming increasingly important [14,15,16,17]. There is an urgent need to identify the risk profile of individual breast cancers—e.g., clinical factors, histopathological factors (TNM classification, tumor grade, hormone receptor status, expression of human epidermal growth factor receptor 2 (HER2), and Ki-67)–and molecular factors that can be used in gene expression analysis panels. There are also biomarkers that have not yet been used in everyday clinical routines for prediction and prognosis, such as cytokeratin 5 (CK5), cluster of differentiation 117 (CD117, c-kit), and epidermal growth factor receptor (EGFR).

Cytokeratins in general belong to the family of intermediate filaments and mark epithelial differentiation. Due to its medium protein size, CK5 is regarded as an intermediate molecular weight cytokeratin, and is also known as a basal cytokeratin due to its expression in basal cells [18]. 

The *KIT* gene, which is located on chromosome 4q12, encodes for CD117, a receptor tyrosine kinase. It is involved in the regulation of cell growth and activation of cell signal cascades, such as proliferation, apoptosis, adhesion, and cell differentiation [19].

The EGFR belongs to another group of receptor tyrosine kinases, specifically the ERBB/HER family of receptor tyrosine kinases, and is also known as human epidermal growth factor receptor 1 (HER1). It activates subsequent molecules such as phosphatidylinositol 3-kinase (PI3K) and the ras/raf/mitogen-activated protein kinase/ERK kinase (MEK)/extracellular signal-regulated kinase (ERK) cascade via heterodimerization [20]. This results in inhibition of apoptosis and expression of genes for proliferation, differentiation, and survival. Increased EGFR activity can thus lead to tumor progression.

Clinical Significance 

The individual clinical significance of CK5, CD117, and EGFR, respectively, has already been investigated. CK5 and CK6 expressions are reported to be statistically significantly associated with both *BRCA1*-(Breast Cancer gene 1)-related breast cancer and a poorer prognosis [21,22]. For TNBC, the expression of CK5/6 and EGFR allows further differentiation into the five-negative phenotype (5NP) and core basal phenotype (CBP) [23,24]. This is clinically relevant since 5NP is associated with a better prognosis [23,25,26,27]. However, the clinical significance of combining all three biomarkers (i.e., CK5, EGFR, CD117) in a panel is as yet unknown.

The aim of our study was to evaluate the prognosis in relation to disease-free survival (DFS) and (OS) in TNBC patients and in patients with other molecular subtypes by assessing the combined immunohistochemical (IHC) expression of CK5, CD117, and EGFR.

## 2. Materials and Methods

### 2.1. Patient Selection

For the scientific issue involved, a cohort of 894 patients participating in the Bavarian Breast Cancer and Controls Study (BBCC) was made available [28]. The BBCC is a breast cancer cohort and controls study and was designed by the University Breast Center for Franconia at the Department of Gynecology and Obstetrics in Erlangen University Hospital in order to identify susceptibility markers and prognostic markers for breast cancer. To record recurrent disease, distant metastases, and death, patients were contacted once a year if the follow-up had not already been carried out by the University Breast Center for Franconia. Mortality data were obtained through specific inquiries at the population registration offices for all patients. 

Patients were eligible for inclusion in the study if they met the following inclusion criteria: aged at least 18 years, diagnosis of invasive, nonmetastatic breast cancer no more than 1 year previously, and there was formalin-fixed, paraffin-embedded (FFPE) tissue available from the primary tumor, in order to create a tissue microarray (TMA). Male patients and patients with metastatic breast cancer, secondary carcinomas, and tumors with missing biomarkers were excluded (Figure 1). 

### 2.2. Clinical Data 

The University Breast Center for Franconia is a breast center certified by the German Cancer Society (Deutsche Krebsgesellschaft) and the German Society for Breast Diseases (Deutsche Gesellschaft für Senologie). Certification was obtained for the purpose of quality control and quality improvement [18,19]. Moreover, treatment procedures are audited annually, requiring treatment in accordance with the German guidelines, for more than 95% of the patients. Numerous epidemiological parameters are integrated into the data collection, all of which can be associated with the risk of developing breast cancer, and are correlated with a questionnaire completed by the patients. Briefly, all clinical and histopathological data were compiled prospectively in an annually audited, certified database. 

### 2.3. Histopathological Assessment

As previously described, corresponding tissue microarrays (TMAs) were created for all patients in the BBCC cohort for whom paraffin blocks were available [29]. As part of the initial documentation of the disease, the tumor status was documented in accordance with the TNM classification. Tumors were graded using the Elston and Ellis method [30]. 

### 2.4. Evaluation of CK5, CD117, and EGFR

Details of all immunohistochemical antibodies and protocols, respectively, are described in the supplement including Appendix A. The evaluation of CK5, CD117, and EGFR immunohistochemistry (IHC) was similarly carried out, taking both the staining intensity and the percentage of stained tumor cells into account. Completely negative staining was classified as score 0, weak positivity or less than 10% positive cells as score 1, more than 10% positive cells and moderate staining as score 2, and more than 10% positive cells and strong staining as score 3 [31,32]. For a more detailed, but at the same time simpler scoring and standardization, we used the same cut-off for all three biomarkers in intensity and percentage. For EGFR and CK5, all results were assessed with a score of 1, 2, or 3 and were considered positive in the context of this analysis, while a score of 0 was rated as negative [31]. For CD117, a score of 0 or 1 was considered negative, and a score of 2 or 3 was considered positive (Appendix A) [32]. IHC was assessed by a pathologist with an expertise in breast cancer pathology and in the assessment of immunohistochemical stainings on TMAs. 

### 2.5. Statistical Analysis

DFS was defined as the time from the date of primary diagnosis to the earliest date of disease progression (distant metastasis, local recurrence, or death from any cause) or the date of censoring. Patients who were lost to follow-up before the maximum observation period of 10 years, or were disease-free after the maximum observation period, were censored at the last date on which they were known to be disease-free or at the maximum observation time. OS was similarly defined.

The primary objective was to assess if a categorization of patients with TNBC by using a panel of the three biomarkers of interest (CK5, CD117, and EGFR) was associated with DFS and OS. For this intention, TNBC patients were divided into two groups (Table 1).

The prognoses for DFS and OS in the group with three biomarkers negative (TBN) and in the group with at least one positive biomarker (non-TBN) were compared using the Kaplan–Meier product limit method and the corresponding log rank test. In addition, 2-year, 5-year, and 10-year survival rates with 95% confidence intervals were calculated for DFS and OS.

The secondary objective was to compare the above-defined groups (TBN vs. non-TBN) in the other three subclasses of molecular breast cancer (Luminal A-like, Luminal B-like, HER2-positive). These were defined as follows: 

Luminal A-like tumors show a positive expression of hormone receptors (estrogen receptor (ER) and/or progesterone receptor (PR)), have a negative HER2 status and are of tumor grade G1 or G2 with Ki-67 under 14%; Luminal B–like tumors are HER2-negative, hormone receptor-positive with grade G3 or Ki-67 of at least 14%. Triple-negative tumors are HER2-negative, ER-negative, and PR-negative. HER2-positive tumors are HER2-positive and ER/PR-positive or ER-/PR-negative [33,34].

Patients for whom survival information was missing or who had missing values for the biomarkers of interest were excluded from the analysis. Missing values for determination of molecular class were replaced as done by Salmen et al. [35]. All of the tests were two-sided, and *p* < 0.05 was regarded as statistically significant. All analyses were carried out using the R system for statistical computing (version 3.6.1; R Development Core Team, Vienna, Austria, 2019).

## 3. Results

### 3.1. Patient Characteristics

Overall, 572 patients were included in this analysis. Of these, 84 patients had a TNBC (14.7%), 212 patients (37.1%) had Luminal A-like, 206 patients (36.0%) had Luminal B-like and 70 patients (12.2%) had HER2+ breast cancer. For the primary objective, 84 TNBC patients were analyzed. Baseline clinical and pathological characteristics for the TNBC subset and the whole study population are shown in Table 2. 

### 3.2. Distribution of CK5, CD117, and EGFR

Overall, the proportions of positivity for CK5, CD117, and EGFR were 10.8%, 11.7%, and 2.4%, respectively (Table 2). In the overall collective, 81.1% were TBN with negativity in CK5, CD117 and EGFR. In contrast to the overall cohort, the positivity rates for CK5, CD117, and EGFR were higher in the TNBC subset with 35.7%, 32.1%, and 9.5%, respectively (Table 2). Comparing the proportions of TBN across the molecular classes, the proportion of tumors expressing none of the three biomarkers was the lowest in TNBC (48.8%) and was much more frequent in Luminal A-like (91.5%), Luminal B-like (86.4%) and HER2-positive (72.9%) lesions. Comparing the proportions of non-TBN (n = 43) and TBN (n = 41) within the TNBC subgroup, non-TBC shows higher grading (G1/2 versus G3) or lower tumor stage (T1 versus T2-4), respectively (Table 3). Figure 2 shows examples of CK5-, CD117-, and EGFR-positive breast cancer cases, respectively.

### 3.3. Prognosis in the TNBC Subgroup

There were significant differences between the two biomarker groups (TBN versus non-TBN) in the TNBC subgroup (n = 84) with regard to DFS (*p* = 0.035) and OS (*p* = 0.022) in the unadjusted survival analysis. The DFS and OS rates were lower in the TBN group (Figure 3 and Figure 4, respectively). The survival rates for 2, 5, and 10 years are shown in Figure 5. Due to a low number of events, the confidence intervals were broad. 

### 3.4. Prognosis in Other Molecular Subclasses

As shown in Figure 5a–f, further division of the other three molecular subclasses (Luminal A-like, Luminal B-like, HER2+) did not show significant differences in the DFS and OS rates between the two biomarker groups (TBN vs. non-TBN).

## 4. Discussion

This retrospective study investigated the prognostic impact of a three-biomarker panel on the outcomes for breast cancer patients with different subtypes of breast cancer. The TBN group (with all three biomarkers CK5, CD117, and EGFR negative) was the lowest in TNBC subgroup. There was a significant difference between the two biomarker groups (TBN versus non-TBN) in the TNBC subgroup. DFS and OS were lower in the TBN subgroup.

The novel aspect of this study is that we examined a three-biomarker combination of CK5, CD117, and EGFR systematically and separately in the four established subgroups of breast cancer, focusing on TNBC.

### 4.1. Prognostic Impact of CK5 in Breast Cancer

One study showed that CK5 positivity is associated with shorter disease-specific survival periods. In the group of node-positive breast cancer patients, the expression of basal markers has been found to be associated with a significantly poorer outcome [27,36]. This study showed that in node-negative breast cancers, those with CK5 expression were associated with significantly poorer survival than those without CK5 expression [37]. Numerous studies have concluded that within TNBC, CK5 is associated with a significantly higher recurrence rate, increased mortality, shorter OS [38,39] and a positive correlation with negative prognostic factors such as lymph-node metastases and a high tumor grade [39,40,41,42,43]. The biomarker group with at least one positive marker for CK5, CD117, or EGFR also had a better prognosis for TNBC. CK5 was the biomarker most commonly expressed in TNBC. In recent studies, the positive expression of CK5 was associated with a better prognosis; its absence, however, had a poorer recurrence-free survival (*p* = 0.02 and 0.002) and OS (*p* = 0.05 and 0.02) in univariate and multivariate analyses [44].

In another study, the recurrence rate among patients with CK5-positive TNBC was lower and survival was higher than among CK5-negative TNBC patients. In combination with E-cadherin, CK5 positivity was associated with the longest EFS (Hazard ratio = 5.075; 95% CI: 1.09–23.53; *p* = 0.038) [45]. Another group of 94 TNBC patients showed a significantly higher expression of CK5 in comparison with the overall group, and CK5 was associated with better OS and DFS (*p* = 0.036) [46]. In another study, the poorest prognosis was found in a subgroup with CK5, androgen receptor negativity, and p53 positivity, regardless of CK5 expression. CK5 did not correlate with other clinical pathological parameters and prognosis markers [47]. The data on the prognostic influence of CK5 in breast cancer patients in the above studies with 52 to 94 TNBC patients were therefore contradictory.

### 4.2. Prognostic Impact of CD117 in Breast Cancer

In a study of 464 patients, CD117 was expressed significantly more often (49% vs. 10%; *p* = 0.001) in the TNBC subgroup (7.3%) in comparison with the non-TNBC group. However, survival analyses have not shown any significant impact of CD117 expression on OS. The expression of other receptor tyrosine kinases, such as platelet-derived growth factor receptor (PDGFR) and vascular endothelial growth factor receptor (VEGFR), has not shown any significant impact on survival data. There were no significant correlations between CD117 and the prognosis [48]. In another study including 930 patients, CD117 expression was more common in basal-like breast cancer or TNBC. However, there was no significant influence on the prognosis [27]. A trial including 190 patients with TNBC showed that CD117-positive TNBC is linked to significantly poorer DFS and cancer-specific survival and that CD117 is an independent prognostic marker for TNBC [16]. The combination of CD117+/*TP53* missense mutation was assumed to be an independent prognostic factor in TNBC and is generally associated with a poorer prognosis [32]. 

Patient age was not considered in this study. In a premenopausal subgroup, no significant differences between TNBC and non-TNBC in regard to amplification of *CD117* were observed using fluorescence in situ hybridization (FISH). Patients who were positive on *CD117* FISH not only showed a correlation with negative ER status and high tumor grade, but also had a poorer survival—but not after the multivariate analysis, when the patients’ age, tumor size, tumor grade, TNBC, and nodal status were taken into account. There were no significant differences in survival in immunohistochemical protein expression of CD117 [49].

In a divergent approach with subgroups depending on nodal status, 14.7% of all breast cancers were found to be positive for CD117. CD117 was significantly higher in distant metastatic and node-positive breast cancers. CD117 was associated with a poorer outcome. The study stated that CD117 is an independent prognostic marker and is prognostically significant, including for the prediction of metastases [50]. Due to discrepancies between the groups studied, direct comparison with this study is challenging.

### 4.3. Prognostic Impact of EGFR in Breast Cancer

The prognostic significance of EGFR is a matter of controversy. A trial including 151 patients with TNBC showed overexpression of EGFR in 64% of TNBC lesions. On FISH assessment, large numbers of gene copies of *EGFR* were found in one-third of cases, and *EGFR* mutations were detected in 3% of TNBCs. This showed that immunohistochemical overexpression of EGFR is significantly associated with amplification and polyploidy on FISH. It also showed that overexpression of EGFR is associated with a lower clinical stage. Analyses of survival rates did not show any association with DFS for *EGFR* mutations or EGFR overexpression. An increased *EGFR* gene copy count was associated with shorter DFS [51].

Another trial observed expression of EGFR in 88.5% of breast cancers. There was no correlation with other clinical and histopathological factors such as tumor grade, lymphatic grade status, or p53 status. Neither overexpression of EGFR nor an increased gene copy count were significantly associated with EFS [52]. Expression of EGFR after neoadjuvant systemic therapy was reported to be significantly reduced. This was considered to show that tumor growth was significantly slower as a result of therapy. Low levels of EGFR expression after treatment correlated with tumor regression and higher survival rates in the study, while high EGFR expression was associated with disease progression, treatment resistance, and sentinel lymph-node metastases [53]. Consequently, it is not the expression of EGFR per se, but rather differences between preoperative and postoperative findings that should be taken into account. In another subgroup analysis, approximately 78% of TNBCs showed a basal-like phenotype, which was defined in the study using positivity of CK5/6 and/or EGFR. These patients were significantly more likely to have lymph-node metastases, larger tumors, more advanced tumor stages, significantly poorer cancer-specific survival, and shorter DFS periods than the other subgroups [31].

In a study by Nielsen et al., in which basal-like breast cancer was defined using a gene expression profile, this subgroup was negative for ER and positive for EGFR and c-KIT. EGFR was expressed in 54% of cases with a basal-like phenotype (in comparison with 11% of cases negative for basal cytokeratins) and was associated with poor survival [27,54,55]. It has also been shown that EGFR is a significant independent prognostic factor [27].

In the present study, however, TNBC was associated with a better prognosis if at least one of the three biomarkers (CK5, CD117, or EGFR) was positive. A possible explanation for this might be found in a study of 198 breast cancers, showing that overexpression of EGFR led to a significantly better response to neoadjuvant chemotherapy and was significantly associated with a higher rate of pathological complete responses. However, a significant association between EGFR and the prognosis was not established [56].

Another explanation might be a different response to chemotherapy. CK5-, CD117-, and EGFR-positive breast cancers might have an increased cell death rate during chemotherapy and consequently a stronger response to chemotherapeutic agents, which could also affect OS and DFS [56,57]

A further possible explanation for the fact that the subgroup with at least one of the markers CK5, CD117, and EGFR being positive was associated with a better DFS and OS prognosis might be the response of TNBC to different chemotherapy regimens based on different biomarkers. In an in vitro experiment, both CD117 and CK5 expression was associated with a significantly higher apoptosis rate to certain chemotherapy agents than biomarker-negative breast cancer [57]. Overall, prognosis determination and pCR prediction in TNBC is challenging so that we have been exploring in our group for years [13,58,59,60]. There are various options, either simple immunohistochemical-based like in this study, more complex and costlier (spatial) multi-OMIC analyses or clinical, such as an inspection or the palpation of the breast, as well as instrumental (ultrasound, mammogram, MRI [61]). None of these aspects work individually, it is rather a combination of as many items as possible to have a more sensitive prognosis prediction. Using the three biomarkers of CK5, CD117 and EGFR can be a valuable additional information on prognosis determination and a feasible option since it is an easily assessable and cost-effective immunohistochemical panel, which can be performed by most pathological laboratories worldwide. 

## 5. Limitations

The evaluation of the three immunohistochemical markers in this study was carried out using quantitative and qualitative evaluation. In direct comparison with the literature, IHC assays and cut-off values for positive categorization vary across studies [27,31,32,47,62]. The cut-off of 10% used in this study was used for CD117 in accordance with Luo et al. [32] and for EGFR and CK5 in Kashiwagi et al. [31]. In our study, this 10% cut-off was chosen for all three biomarkers for a more detailed and unified result. 

In the presented literature, there are conflicting results regarding the impact of the three biomarkers on the prognosis of breast cancer patients. Thus, this simple panel is of limited use in clinical routine.

Further research should aim for standardized evaluation of these parameters. Differences in preanalytical and analytical methods, including different antibody clones as well as differences in reading the IHC stains, may lead to varying results in evaluating marker expression. In our study, we have assessed both staining intensity and the percentage of stained tumor cells in accordance with referenced literature [22,27,31,32,63], since a two variable score reflects the variable expression better than by using percentage alone (e.g., H-score, immunoreactive score according to Remmele and Stegner) [64]. A detailed procedural guide is essential to ensure transparency and comparability. 

Our results, however, should be interpreted with caution due to the low number of cases, especially the small caseload of TNBC. This is caused by using a cohort with all the molecular subtypes for creating the TMA; the focus was not on specifically choosing TNBC. The drop-outs of IHC assessment using TMAs are due to some cores not showing a sufficient number of tumor cells or cores having been washed away during the IHC staining protocol. Therefore, statistical analysis of TMA biomarker evaluations usually consists of fewer cases than the initial TMA cohort [29,33,65]. This cohort has nevertheless been chosen because of its excellent clinical characterization and long follow-up data.

There are also biological limitations. In particular, CK5 can be expressed heterogeneously, so that evaluation via TMA might lead to false-negative results [66]. In addition, the study was conducted retrospectively on archived tumor tissue.

## 6. Conclusions

Immunohistochemical analysis of the prognostic value of CK5, CD117, and EGFR showed a longer DFS and OS periods for TNBC patients (with the tumor expressing at least one positive biomarker CK5, CD117 and/or EGFR) in contrast to the findings reported in other studies. 

TNBC is a molecular subtype of breast cancer associated with a poor prognosis and a good response to chemotherapy. It is therefore essential not only to identify predictive biomarkers, but also biomarker combinations that can help to divide TNBCs into subgroups with different prognoses. By using this three-biomarker panel, which can easily be implemented in daily routine diagnostics, the use of individualized risk profiles for patients with TNBC can be improved. Moreover, an interesting and promising step on this path might be the combination of conventional immunohistochemical markers with multigene analysis in order to define an individualized prognostic profile.

## Figures and Tables

**Figure 1 diagnostics-13-00372-f001:**
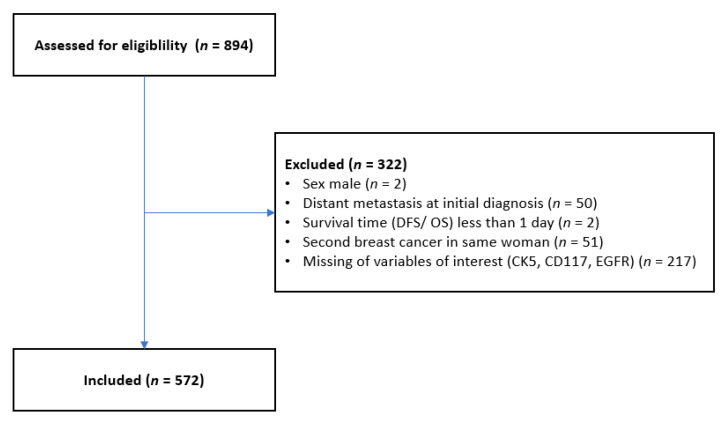
Patient selection and exclusion criteria in this study. DFS, disease-free survival; OS, overall survival.

**Figure 2 diagnostics-13-00372-f002:**
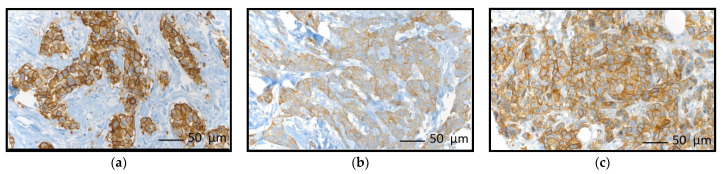
Breast cancer cases with positive biomarker status for (**a**) EGFR, (**b**) CK5, and (**c**) CD117 immunohistochemistry, respectively (each 400× magnification).

**Figure 3 diagnostics-13-00372-f003:**
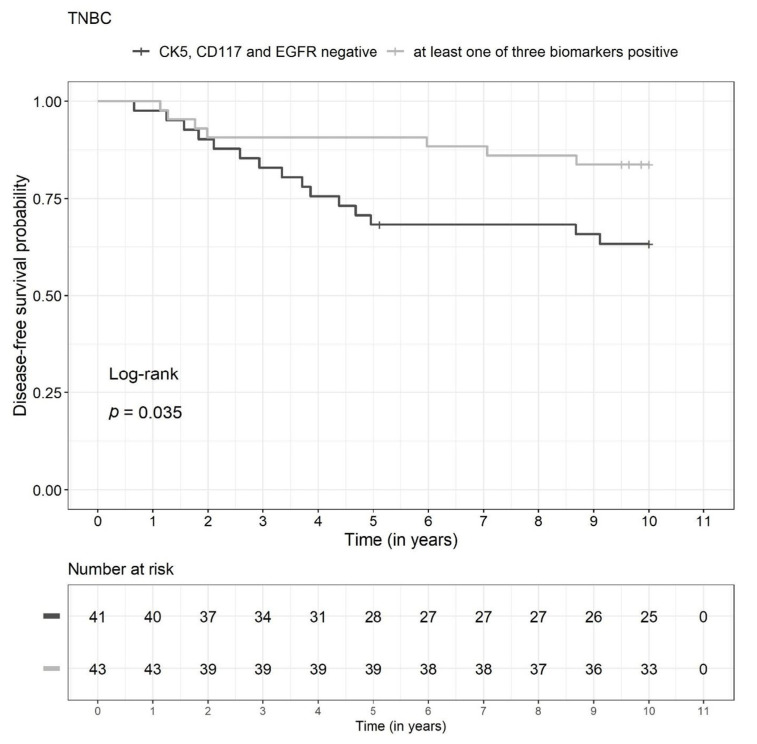
Kaplan-Meier curves for all three biomarkers negative (TBN) versus at least one biomarker positive (non-TBN) in the triple-negative breast cancer (TNBC) subgroup (n = 84) relative to disease-free survival.

**Figure 4 diagnostics-13-00372-f004:**
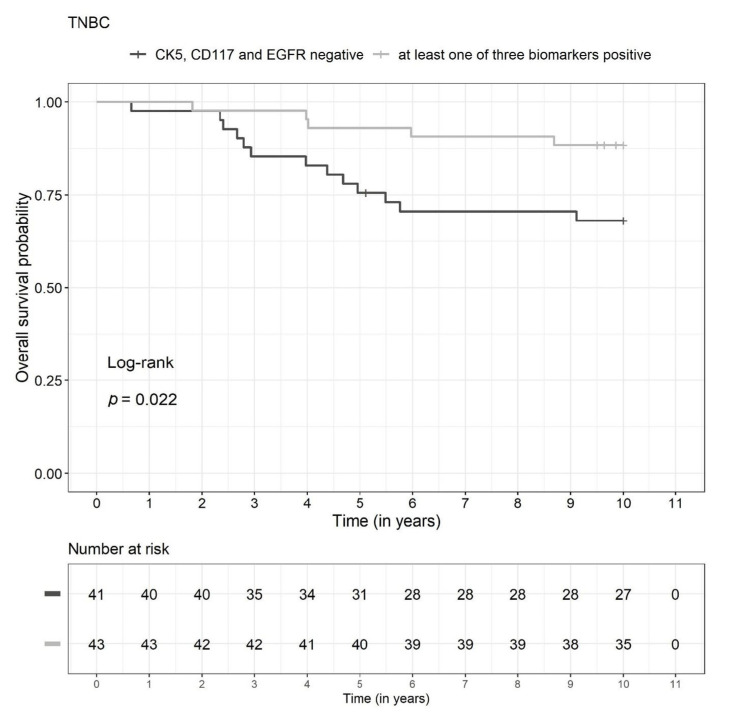
Kaplan-Meier curves for all three biomarkers negative (TBN) versus at least one biomarker positive (non-TBN) in the triple-negative breast cancer (TNBC) subgroup (n = 84) relative to overall survival.

**Figure 5 diagnostics-13-00372-f005:**
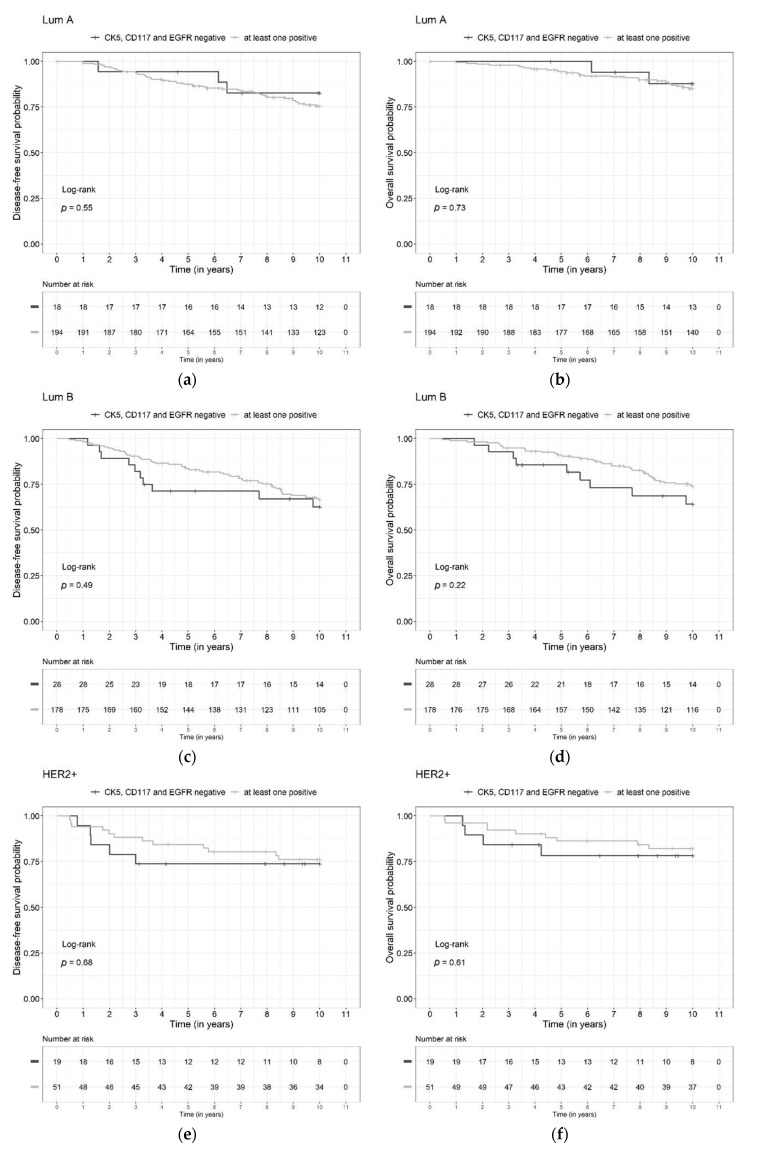
Kaplan-Meier curves for all three biomarkers negative (TBN) versus at least one biomarker positive (non-TBN) in Luminal A-like (**a**,**b**), Luminal B-like (**c**,**d**), and HER2-positive (**e**,**f**) relative to disease-free (**a**,**c**,**e**) and overall survival (**b**,**d**,**f**).

**Table 1 diagnostics-13-00372-t001:** Division of TNBC patients into TBN and Non-TBN.

Subgroup	CK5	CD117	EGFR
TBN	0 and	0 or 1 and	0
Non-TBN	>0 and/or	>1 and/or	>0

CD117 cluster of differentiation 117; CK5 cytokeratin 5; EGFR epidermal growth factor receptor; TBN three biomarkers negative; TNBC triple-negative breast cancer.

**Table 2 diagnostics-13-00372-t002:** Baseline characteristics of clinical and pathological parameters for patients with triple-negative breast cancer (n = 84) and all patients (n = 572).

Parameter	Category	TNBC	Total
Patients	n	84	572
Age	Mean (SD)	55.8 (13.2)	57.5 (12.3)
BMI	Mean (SD)	26.2 (4.9)	26.4 (5.2)
Ki-67	Mean (SD)	43.1 (25.7)	24.0 (20.9)
ER	Positive	0 (0)	448 (78.3)
PR	Positive	0 (0)	419 (73.3)
HER2	Positive	0 (0)	70 (12.2)
Tumor grade (G)	G1	1 (1.2)	45 (7.9)
	G2	31 (36.9)	386 (67.5)
	G3	52 (61.9)	141 (24.7)
Nodal status (N)	N+	36 (42.9)	243 (42.5)
Tumor stage (T)	T1	43 (51.2)	294 (51.4)
	T2	35 (41.7)	227 (39.7)
	T3	3 (3.6)	28 (4.9)
	T4	3 (3.6)	23 (4.0)
Molecular subtype	TNBC	84 (100)	84 (14.7)
	Luminal A-like	0 (0)	212 (37.1)
	Luminal B-like	0 (0)	206 (36.0)
	HER2+	0 (0)	70 (12.2)
CK5	Positive	30 (35.7)	62 (10.8)
CD117	Positive	27 (32.1)	67 (11.7)
EGFR	Positive	8 (9.5)	14 (2.4)

Values are given as n (%) unless otherwise specified. BMI body mass index, EGFR epidermal growth factor receptor, ER estrogen receptor, IQR interquartile range, PR progesterone receptor, TNBC triple-negative breast cancer.

**Table 3 diagnostics-13-00372-t003:** Baseline characteristics of clinical and pathological parameters for patients with all triple-negative breast cancer into non-TBN (n = 43) and TBN (n = 41).

Parameter	Category	Non-TBN	TBN
Patients	n	43	41
Age	Mean (SD)	52.7 (14.2)	59.0 (11.3)
BMI	Mean (SD)	25.7 (5.2)	26.7 (4.6)
Ki-67	Mean (SD)	52.6 (25.0)	33.3 (22.7)
Tumor grade (G)	G1	0	1 (2.4)
	G2	10 (23.3)	21 (51.2)
	G3	33 (76.7)	19 (46.3)
Nodal status (N)	N+	16 (37.2)	20 (48.8)
Tumor stage (T)	T1	27 (62.8)	16 (39.0)
	T2	15 (34.9)	20 (48.8)
	T3	1 (2.3)	2 (4.9)
	T4	0 (0.0)	3 (7.3)
CK5	Negative	13 (30.2)	41 (100)
	Positive	30 (69.8)	0
CD117	Negative	16 (37.2)	41 (100)
	Positive	27 (62.8)	0
EGFR	Negative	35 (81.4)	41 (100)
	Positive	8 (18.6)	0

Values are given as n (%) unless otherwise specified. BMI body mass index, EGFR epidermal growth factor receptor, IQR interquartile range, non-TBN one or more of three biomarkers (EGFR, CK5, CD117) positive, TBN three biomarkers (EGFR, CK5, CD117) negative.

## Data Availability

The datasets are available from the corresponding author on reasonable request.

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
