# Peer review of "Expression of the Immunohistochemical Markers CK5, CD117, and EGFR in Molecular Subtypes of Breast Cancer Correlated with Prognosis"

_diagnostics, 2023, doi:10.3390/diagnostics13030372_

Round 1

Reviewer 1 Report

To identify effective and routinely implemented tests to subdivide TNBC into prognosis-specific groups is extremely actual and significant issue for breast pathology and mammalogy. Authors demonstrated a prognostic IHC panel with CK 5, CD 117 and EGFR. These markers are used for many tumors diagnostics in routine practice, thus, many IHC laboratories are supplied with them and it is a positive characteristic for these panel. At the same time authors mentioned that it is impossible to use them separately because none of these markers did not demonstrate rather high specificity. Moreover, the results of the other investigations tested these markers showed controversial results concerning CK 5, CD 117 and EGFR power to subdivide TNBC into good-prognosis and poor-prognosis groups. This is a disadvantage of this panel. In addition, the authors do not discuss other contemporary methods for prognosis determination in TNBC patients either ICH-based or others (OMICs-based, clinical, instrumental etc). It should be added from my point of view to demonstrate the pluses of the proposed panel and convince the readers to use it instead of others methods.

Author Response

Expression of the immunohistochemical markers CK5, CD117 and EGFR in molecular subtypes of breast cancer in correlation to prognosis (Ref: Manuscript ID: diagnostics-2140128)

Revision_1- Reply to reviewer 1

Dear Editors,

Dear reviewer,

Dear ladies and gentlemen,

We are grateful for your comprehensive review and we appreciate your effort in time and your instructive and helpful comments. We have revised our manuscript according to your instructions. We hope you are satisfied with the changes of our manuscript. If further revisions should be requested, we will do so. Please find below our point-for-point replies to your comments.

Yours sincerely,

Paul Gaß on behalf of the authors

Revision_1 - Reply to reviewers

Reviewer 1

To identify effective and routinely implemented tests to subdivide TNBC into prognosis-specific groups is extremely actual and significant issue for breast pathology and mammalogy. Authors demonstrated a prognostic IHC panel with CK 5, CD 117 and EGFR.

Reviewer’s comment

Authors’ reply

1. These markers are used for many tumors diagnostics in routine practice, thus, many IHC laboratories are supplied with them and it is a positive characteristic for these panel.

Thank you very much for pointing out the feasibility of using this panel. This important aspect has been added to the discussion (page 15, lines 373-377).

2. At the same time authors mentioned that it is impossible to use them separately because none of these markers did not demonstrate rather high specificity. Moreover, the results of the other investigations tested these markers showed controversial results concerning CK 5, CD 117 and EGFR power to subdivide TNBC into good-prognosis and poor-prognosis groups. This is a disadvantage of this panel. 

We fully comprehend this point of criticism and have added this issue to the limitations (page 15, lines 386-388).

3.   In addition, the authors do not discuss other contemporary methods for prognosis determination in TNBC patients either IHC-based or others (OMICs-based, clinical, instrumental etc). It should be added from my point of view to demonstrate the pluses of the proposed panel and convince the readers to use it instead of others methods.

Thank you for your valuable input, other aspects of prognosis determination have been added to the discussion (page 15, lines 367-377). It is most likely a combination of as many methods as possible in order to predict prognosis in breast cancer and especially in TNBC.

Reviewer 2 Report

Dear Authors:

The manuscript "Expression of the immunohistochemical markers CK5, CD117, and EGFR in molecular subtypes of breast cancer correlated with prognosis" by Schulmeyer et al has demonstrated that the non-TBN subgroup of TNBC lesions with at least one positive biomarker of CK5, CD117, and/or EGFR, to be associated with longer DFS and OS. I have just a few suggestions.

1.  Some references or information are missing.

In introduction, please add more background information about breast cancer, which can emphasize the importance of your article. (please cite: 1. Advances in the Prevention and Treatment of Obesity-Driven Effects in Breast Cancers. Front Oncol. 2022 Jun 22;12:820968. doi: 10.3389/fonc.2022.820968. PMID: 35814391; PMCID: PMC9258420.

2. An Epigenetic Role of Mitochondria in Cancer. Cells. 2022 Aug 13;11(16):2518. doi: 10.3390/cells11162518. PMID: 36010594; PMCID: PMC9406960.

3. Mitochondrial mutations and mitoepigenetics: Focus on regulation of oxidative stress-induced responses in breast cancers. Semin Cancer Biol. 2022 Aug;83:556-569. doi: 10.1016/j.semcancer.2020.09.012. Epub 2020 Oct 6. Erratum in: Semin Cancer Biol. 2022 Jul 16;: PMID: 33035656.)

Best,

Author Response

Expression of the immunohistochemical markers CK5, CD117 and EGFR in molecular subtypes of breast cancer in correlation to prognosis (Ref: Manuscript ID: diagnostics-2140128)

Revision_1- Reply to reviewer 2

Dear Editors,

Dear reviewer,

Dear ladies and gentlemen,

We are grateful for your comprehensive review and we appreciate your effort in time and your instructive and helpful comments. We have revised our manuscript according to your instructions. We hope you are satisfied with the changes of our manuscript. If further revisions should be requested, we will do so. Please find below our point-for-point replies to your comments.

Yours sincerely,

Paul Gaß on behalf of the authors

Revision_1 - Reply to reviewers

Reviewer 2

Dear Authors:

The manuscript "Expression of the immunohistochemical markers CK5, CD117, and EGFR in molecular subtypes of breast cancer correlated with prognosis" by Schulmeyer et al has demonstrated that the non-TBN subgroup of TNBC lesions with at least one positive biomarker of CK5, CD117, and/or EGFR, to be associated with longer DFS and OS. I have just a few suggestions.

Reviewer’s comment

Authors’ reply

1.  Some references or information are missing.

In introduction, please add more background information about breast cancer, which can emphasize the importance of your article. (please cite: 1. Advances in the Prevention and Treatment of Obesity-Driven Effects in Breast Cancers. Front Oncol. 2022 Jun 22;12:820968. doi: 10.3389/fonc.2022.820968. PMID: 35814391; PMCID: PMC9258420.

2. An Epigenetic Role of Mitochondria in Cancer. Cells. 2022 Aug 13;11(16):2518. doi: 10.3390/cells11162518. PMID: 36010594; PMCID: PMC9406960.

3. Mitochondrial mutations and mitoepigenetics: Focus on regulation of oxidative stress-induced responses in breast cancers. Semin Cancer Biol. 2022 Aug;83:556-569. doi: 10.1016/j.semcancer.2020.09.012. Epub 2020 Oct 6. Erratum in: Semin Cancer Biol. 2022 Jul 16;: PMID: 33035656.)

Thank you very much for pointing out that more background information on breast cancer in the introduction is necessary.

A part of references have been added to the introduction to offer a broader understanding of breast cancer with their role of epigenetic and an individualized diagnostics and treatment (page 2, lines 68-70).

Reviewer 3 Report

see the report

Author Response

Expression of the immunohistochemical markers CK5, CD117 and EGFR in molecular subtypes of breast cancer in correlation to prognosis (Ref: Manuscript ID: diagnostics-2140128)

Revision_1- Reply to reviewer 3

Dear Editors,

Dear reviewer,

Dear ladies and gentlemen,

We are grateful for your comprehensive review and we appreciate your effort in time and your instructive and helpful comments. We have revised our manuscript according to your instructions. We hope you are satisfied with the changes of our manuscript. If further revisions should be requested, we will do so. Please find below our point-for-point replies to your comments.

Yours sincerely,

Paul Gaß on behalf of the authors

Revision_1 - Reply to reviewers

Reviewer 3

The paper targets the use of individualized risk profiles for patients with triple negative breast cancer. A detailed presentation of the cohort is followed by the description of the proposed method. A comparison of disease free survival and overall survival supports the proposed interesting and promising step in the combination of conventional immunohistochemical markers with multigene analysis. The paper might be suitable for publication in Diagnostics. However, the following requirements have to be fulfilled.

Reviewer’s comment

Authors’ reply

1. in lines 50-57 you should include the reference

·         Robustness Evaluation of a Deep Learning Model on Sagittal and Axial Breast DCE-MRIs to Predict Pathological Complete Response to Neoadjuvant Chemotherapy, Journal of Personalized Medicine (2022);

Thank you very much for your valuable input on personalized medicine and individualized prediction of pCR. This reference has been added to the Introduction (page 2, lines 56-58).

2. in line 53 you have to specify how an event-free survival is characterized in a different way with respect to disease-free survival;

Thank you for your important remark, the definition of event-free survival has been added to line 53 to provide a clearer distinction to disease-free survival. 

3. in lines 58-67 you should include the reference ˆ A roadmap towards Breast Cancer Therapies Supported by Explainable Artificial Intelligence, Applied Sciences (2021);

Thank you very much for your valuable information  on personalized medicine and clustering through artificial intelligence. This quotation has been added to the Introduction (page 2, lines 56-60). .

4. in line 72 you have to delete ”, is,”

Thank you very much for your comment, “is” has been deleted.

5. in lines 156-158 and 172-176 you should increase the fontsize

Thank you very much for your diligent comment, the frontsize has been adapted.

6. in Figure 2 the panels can be arranged in just one row;

The panels of Figure 2 have been rearranged into one row.

7. in line 210 you to refer to ”Figures 3 and 4”

Thank you very much for your comment, the referring to Figures 3 and 4 has been changed.

8. in line 211 you have to refer to ”Figure 5”

Thank you very much for your comment, the referring to Figure 5 has been changed.

9. in Figure 5 an arrangement of panels according to rows associated with LumA, LumB and HER2 is more clear;

Thank you for your valuable input, the panels in Figure 5 have been rearranged.

10. in lines 262-263 the statement ”The data on the prognostic influence of CK5 in breast cancer patients are therefore contradictory” has to include a reference about the number of patients adopted in the considered study to motivate the contradictions related with a low statistucs

Thank you very much for this remark as this is a very important issue. The amount of TNBC in the referred studies differs from 52 to 94 patients. This has been added as following: “The data on the prognostic influence of CK5 in breast cancer patients in the above studies with 52 to 94 TNBC patients are therefore contradictory.”(page 13 lines289-291)

11. in line 363 you have to use a capital letter ”Immuno...”.

Thank you very much, the word has been corrected.